# Different Approaches in Therapy Aiming to Stabilize an Unstable Atherosclerotic Plaque

**DOI:** 10.3390/ijms22094354

**Published:** 2021-04-21

**Authors:** Michal Kowara, Agnieszka Cudnoch-Jedrzejewska

**Affiliations:** Laboratory of Centre for Preclinical Research, Department of Experimental and Clinical Physiology, Medical University of Warsaw, 02-091 Warsaw, Poland; michal.kowara@wum.edu.pl

**Keywords:** atherosclerotic plaque, stabilization, vulnerable plaque, inflammation, macrophage, intravascular ultrasound, myocardial infarction, necrotic cores, cell death

## Abstract

Atherosclerotic plaque vulnerability is a vital clinical problem as vulnerable plaques tend to rupture, which results in atherosclerosis complications—myocardial infarctions and subsequent cardiovascular deaths. Therefore, methods aiming to stabilize such plaques are in great demand. In this brief review, the idea of atherosclerotic plaque stabilization and five main approaches—towards the regulation of metabolism, macrophages and cellular death, inflammation, reactive oxygen species, and extracellular matrix remodeling have been presented. Moreover, apart from classical approaches (targeted at the general mechanisms of plaque destabilization), there are also alternative approaches targeted either at certain plaques which have just become vulnerable or targeted at the minimization of the consequences of atherosclerotic plaque erosion or rupture. These alternative approaches have also been briefly mentioned in this review.

## 1. Introduction

Atherosclerotic plaque is the pathophysiological basis of ischemic heart disease–a widespread disease in both developed and developing countries [1]. A principal complication of ischemic heart disease is myocardial infarction, which constitutes the main cause of mortality worldwide [2]. Myocardial infarction is a consequence of the sudden occlusion of the coronary artery supplying the myocardial tissue, which is caused mainly (75% of the time) by atherosclerotic plaque rupture and thrombus generation [3,4,5]. This deficiency in the oxygen supply leads to myocardial cell damage and necrosis which compromises the entire heart activity and might result in ion current disturbances leading to life-threatening ventricular arrhythmias [6,7]. The development of atherosclerotic plaque is a long process composed of several stages (Figure 1).

Briefly, apo-B containing lipoproteins (especially LDL) pass through the endothelial layer of the arterial intima and accumulate within the subendothelial area where they are endocytosed by intimal macrophages. Simultaneously, local blood flow disturbances (non-linear flow) in atherosclerosis-susceptible regions (e.g., arterial branches) cause decreased shear stress that is detected by the endothelial cells in the process of mechanotransduction. These processes change the microenvironment of the arterial wall intima, which promotes subsequent alterations—foam cell generation, vascular smooth muscle cell migration, and their conversion from contractile into the synthetic phenotype, extracellular matrix remodeling, plaque growth, fibrous cap formation, and finally, necrotic core formation and calcifications [8,9]. In addition, Peter Libby emphasized the crucial role of the immune system in atherogenesis [10]. Many plaques develop into stable structures which manifest clinically as chronic coronary syndrome, but some of them undergo special ultrastructural alterations making them prone to rupture. Such plaques are called ‘unstable’ or ‘vulnerable’ [11]. According to the classic definition created by Virmani, vulnerable plaque is a thin-cap fibroatheroma (TCFA), characterized by a necrotic core presence with an overlying fibrous cap of thickness <65 μm [12]. The processes leading to plaque destabilization, i.e., apoptosis of VSMC within the fibrous cap, neovascularization, and necrotic core enlargement, are presented in Figure 2 [13].

The occurrence of major cardiovascular adverse events (MACE—composite endpoint composed of death from cardiac causes, cardiac arrest, myocardial infarction, or rehospitalization because of unstable or progressive angina) relies on the presence of vulnerable plaques. A PROSPECT study on 697 patients with acute coronary syndrome treated with PCI revealed that the cumulative rate of MACE during a three-year follow-up referred to untreated, non-culprit lesions was significantly and independently correlated with the presence of thin-cap fibroatheroma on intravascular ultrasonography [14]. Therefore, therapies aimed at promoting atherosclerotic plaque stabilization would be warranted. However, because thin-cap fibroatheromas and stable fibroatheromas can transform into one another and the occurrence of vulnerable atherosclerotic plaque might be underestimated in clinical studies due to clinical silence, the optimal therapy aimed at stabilizing plaques should be concentrated on the promotion of molecular stabilizing pathways ‘in general’ rather than on the stabilization of certain atherosclerotic lesions (considered vulnerable on imaging) [15]. In this review, we summarize the crucial approaches towards plaque stabilization, and we present the clinical studies based on them.

## 2. General Considerations

Antiatherosclerotic properties are important features of certain drugs or chemical compounds. Indeed, a plethora of different substances—from precisely targeted molecules to traditional Chinese medicinal herbs, nutrients, or even gases such as hydrogen have been extensively investigated in search of their potentially beneficial impact on atherosclerotic plaques [16,17,18,19]. The animal models widely used in such investigations are genetically modified (ApoE−/− or LDL−/−) mice as well as WHHL (Watanabe-heritable hyperlipidemic) rabbits [20,21]. In such investigations, a vulnerable atherosclerotic plaque is obtained through dietary modification (i.e., high-fat diets containing high cholesterol) together with interventions such as angiotensin II infusion via an osmotic pump, cast placement around the common carotid artery, or balloon injury (in rabbits) [22,23]. Then, the animals are divided into a control group (receiving a saline solution) and the experimental group (receiving the study compound). When the study is finished, the vulnerable plaque indicators are assessed by immunohistochemical analysis in both animal groups. Although many preclinically investigated compounds have presented antiatherosclerotic features, only a minority of them have been proven to be clinically relevant. This situation occurs not only due to dissimilarities between the pathophysiology of atherosclerotic plaque development in animals and humans but also because of the difficulties in the design and performance of applicable clinical studies. Nevertheless, experiments with animal models make it possible to investigate different approaches towards plaque stabilization.

## 3. Approaches Directed at Specific Molecular Pathways

### 3.1. Approach towards Regulation of Metabolism

Knowledge about vulnerable atherosclerotic plaque development makes it possible to investigate whether therapies targeted at specific molecular pathways are able to stabilize it. One kind of such therapy is directed towards the modulation of metabolism. Oxidized low-density lipoproteins are considered to be crucial elements in the process of plaque initiation and progression; therefore, interventions aiming to decrease their level are considered plaque-stabilizing. Lipid-lowering agents, such as statins (hydroxymethylglutaryl-CoA synthase, i.e., cholesterol-synthesizing enzyme inhibitors), ezetimibe (an inhibitor of intestinal cholesterol absorption and Niemann-Pick C1-Like 1 antagonist), and alirocumab (antibody blocking proprotein convertase subtilisin/kexin type 9 and increasing LDL–LDLR recycling) improved atherosclerotic plaque stability in different animal models [21,24,25,26]. Contrary to the LDL particles, high-density lipoproteins (HDL) are considered atheroprotective because of reverse cholesterol transport promotion from lesional macrophages to the liver via interaction with the ATP-binding cassette transporter ABCA1 [27]. It has been demonstrated that recombinant HDL particles (especially Milano type rather than wild-type) promote plaque stability by decreasing intraplaque MMP-2 activity and the chemokine MCP-1 level when compared with a placebo in atherosclerotic New Zealand White rabbits [28]. The cholesteryl ester transfer protein (CETP) inhibitor anacetrapib, which increases HDL levels in serum, also presented similar antiatherosclerotic properties in mice [29]. However, even preclinical studies have demonstrated that LDL level reduction is more important in atherosclerotic plaque stabilization than an increase in HDL [30]. Apart from the lipids, glucose at higher concentrations also participates in the destabilization of atherosclerotic plaque. A study on diabetic ApoE−/− mice showed that hyperglycemia destabilizes the plaque via the inhibition of AMPKα and its target gene, prolyl-4-hydroxylase alpha 1 (P4Hα1), an enzyme participating in collagen synthesis [31]. In consequence, hypoglycemic medications are supposed to be plaque-stabilizing and are therefore antiatherosclerotic agents.

### 3.2. Approach towards Macrophages and Cellular Death Mechanisms

Macrophages contribute significantly to the process of atherosclerotic plaque destabilization. On the one hand, they orchestrate different, sometimes opposite, reactions and molecular pathways within the plaque microenvironment, promoting either plaque instability (M1 subpopulation) or plaque stability (M2 subpopulation) [32]. On the other hand, their conversion into lipid-laden foam cells and subsequent cellular death (in particular through necrosis or necroptosis, i.e., programmed necrosis) lead directly to necrotic core enlargement and plaque destabilization [33]. The situation is different when macrophages undergo typical programmed cellular death, i.e., apoptosis, which can be even atheroprotective in the initial phase of plaque development [34]. However, for plaque stability, it is necessary that the apoptotic bodies are robustly cleared through efferocytosis [35]. Another mechanism of cellular death that prevents plaque destabilization is autophagy, in which damaged organelles or cellular compartments are sequestrated into double-membrane structures called autophagosomes and then degraded by lysosomes [36]. Therefore, molecular pathways preventing macrophage conversion into foam cells or necrosis, as well as pathways promoting autophagy and appropriate clearance of apoptotic bodies (efferocytosis), may be atheroprotective. A study has demonstrated that ApoE−/− mice receiving arglabine, an NLRP3 antagonist which redirects macrophages towards autophagic pathways, presented a decreased level of IL-1β (a marker of inflammation) in plasma and reduced atherosclerotic lesions when compared with ApoE−/− mice from the control group [37]. For appropriate efferocytosis, a Mer Tyrosine Kinase (MerTK) expressed on the macrophage surface is required and pathways, which promote MerTK shedding from a membrane to the soluble form, impair this process. For instance, angiotensin II negatively affects efferocytosis through ADAM17 activation and subsequent MerTK shedding [38]. Apart from the aspects of cellular death, interventions preventing macrophage conversion into foam cells (such as inhibition of LOX-1, a receptor recognizing and internalizing oxLDL particles) as well as reverse cholesterol transport promotion in foam cells via LXRα receptor activation have been demonstrated to be atheroprotective in animal models [39,40,41].

### 3.3. Approach toward Inflammation and Immune Reactions

Vulnerable atherosclerotic plaques are characterized by a robust infiltration of different immune cells. Activation of lesional macrophages causes the production and secretion of diverse interleukins and chemokines, which subsequently drive immune cell infiltration [42,43]. Chemokines recruit neutrophils (CCL2) and T cells (CX3CL1, in more advanced lesions), whereas interleukins, such as IL-1β, activate those cells. Moreover, the immune cells transport through the endothelial layer (diapedesis) depend on adhesive molecules, such as ICAM-1 and VCAM-1. Significant augmentation of T cells (especially CD8+, i.e., cytotoxic T cells) has been observed within atherosclerotic plaque in vulnerable plaque specimens (derived from a biobank of human aortas covering the full spectrum of atherosclerotic disease) [44]. Therefore, antagonizing immune reactions responsible for the inflammatory state within the plaque would be an interesting therapeutic option [45]. As anticipated, inhibition of proinflammatory interleukins (such as IL-6) and chemokines (such as CXCL10) results in atherosclerotic plaque stabilization [46,47]. Interestingly, IL-1β inhibition over the entire time of plaque development in a mouse model resulted in a reduction in atherosclerotic plaque formation, but when such therapy was applied in mice with advanced lesions (from 18 to 24 weeks), not only was it not atheroprotective, but it also resulted in an increased number of macrophages within the plaque and abrogated beneficial remodeling [48,49]. This means that the entire system of interactions and reciprocal feedbacks during atherosclerotic plaque destabilization is very complex, and simple approaches (i.e., blocking factors that are considered to be proinflammatory) tend to be insufficient. For this reason, more general anti-inflammatory approaches are also used, for instance, colchicine, which has presented complex effects by inhibiting critical inflammatory signaling networks (inflammasome, proinflammatory cytokines, and adhesion molecules) [50,51]. Moreover, inflammatory mediators such as interleukins are under the control of transcription factors. For instance, an NF-кB transcriptional factor is considered to be a crucial element of the inflammatory response, and many plaque-stabilizing effects induced by different substances are accompanied by NF-кB suppression [52,53,54]. However, inflammatory pathways are also under the negative control of anti-inflammatory cytokines, and the promotion of anti-inflammatory pathways (e.g., genetic amplification of IL-37) presents a stabilizing effect on the plaque [55]. The cells responsible for the resolution of immune reactions are regulatory T cells (Tregs). It has been demonstrated in the ApoE−/− mouse model that pioglitazone (an antidiabetic drug belonging to the thiazolidinedione group) stabilized the atherosclerotic plaque, which was accompanied by an increase in the number of Tregs within the lesion [56]. Moreover, there is the innovative idea of using vaccines against atherosclerosis. In contrast to vaccines against infectious agents, vaccines against atherosclerosis aim to induce immune tolerance towards core antigens involved in atherosclerotic plaque development (such as ApoB-100 on oxLDL particles). The principal way to gain tolerance is Treg induction through the injections of peptides—fragments of the target antigen. An additional way for such vaccination is the generation of neutralizing autoantibodies, for instance, against PCSK9, which demonstrate antiatherosclerotic properties. These effects can be obtained with the use of special techniques such as the use of adjuvants, neoepitope development technologies, and vaccine platforms (e.g., Qβ bacteriophage virus-like particles) [57].

### 3.4. Approach towards Reactive Oxygen Species—Antioxidation Therapy

Reactive oxygen species play a pivotal role in atherosclerotic plaque progression and subsequent destabilization. They are generated mainly by enzymes, such as NADPH oxidases (NOX) localized within endothelial cells, fibroblasts, and vascular smooth muscle cells, and their expression is upregulated by proinflammatory factors such as IL-1β or Ang II [58,59]. First, reactive oxygen species cause lipid oxidation, which generates oxidized cholesterol derivatives such as 7-ketocholesterol (7-K) and 7β hydroxycholesterol (7β-OH). These compounds insert themselves into the cellular membrane and mediate a plethora of subsequent pathways inducing endothelial pump dysfunction, cell cycle blockade, and lysosomal or endoplasmic reticulum membrane damage in different cells (especially macrophages), leading to their apoptosis [60]. Second, reactive oxygen species cause direct damage to the DNA and promote mechanisms of cellular death in that way [61]. Moreover, oxidized cholesterol derivatives (27-hydroxycholesterol and aldehyde 4-hydroxynonenal) increase prostaglandin E production, which further enhances proinflammatory cytokines and matrix-degrading enzymes (especially matrix metalloproteinase 9), increasing the risk of atherosclerotic plaque rupture [62]. Reactive oxygen species are reciprocally linked with immune reactions. On the one hand, their production is under the control of proinflammatory cytokines such as TNF-α. On the other hand, the products of ROS activity, such as oxidized cholesterol derivatives, promote the expression of proinflammatory cytokines, e.g., IL-1β, TNF-α, IL-8, or chemokine MCP-1, as has been demonstrated in ApoE−/− mice lacking TNF-α and the human monocytic cell lines (U937 and THP-1) [63,64]. It is clear that the overproduction of reactive oxygen species is detrimental for the cells within the atherosclerotic plaque, and pathways induced by these agents may destabilize the plaque, leading to its rupture [65]. Therefore, pharmacological interventions leading to the neutralization of reactive oxygen species (either directly, in different mechanisms of antioxidation, or indirectly, by the abrogation of their generation) would be beneficial in atherosclerotic plaque stabilization. Interestingly, antioxidants are important food components, which leads to the idea that diet can be atheroprotective and plaque-stabilizing in that way. Indeed, many preclinical investigations on different nutrients have proven their stabilizing potential upon atherosclerotic plaque (such as soy isoflavones, vitamin E, carotenes, or xanthines) [66,67,68,69]. For example, ApoE−/− mice fed on blackberry extract rich with anthocyanin presented increased HDL levels and increased connective tissue content within the plaque, resulting in improved plaque stability when compared with the control group [70]. Moreover, the effect of stabilizing atherosclerotic plaques was also achieved by action on the controlling mechanisms; NOX2 (NADPH oxidase isoform 2, ROS generator) inhibition resulted in plaque stabilization, whereas Hsp70 (a protective chaperone) silencing resulted in plaque destabilization [71,72].

### 3.5. Approach towards Extracellular Matrix Remodeling and Neovascularization

It has to be emphasized that the entire plaque structure is based on scaffolding, i.e., an extracellular matrix (ECM) composed of collagens, elastin, proteoglycans, and fibronectin. These proteins constitute the connective tissue of the plaque. A fibrous cap, i.e., structure covering the plaque and maintaining its stability, is built up with vascular smooth muscle cells and macrophages embedded in collagen and elastin fibers. The appropriate, compact, and organized structure of the ECM component is crucial for plaque stability. Therefore, the influence on ECM protein synthesis or regulation by different methods (including miRNA) affects plaque stability [73,74,75]. The ECM structure is regulated by proteases—cathepsins, serine proteases, and metalloproteinases—matrix metalloproteinases (MMPs), α-disintegrin, metalloproteinases (ADAMs), and α-disintegrin and metalloproteinases with thrombospondin domains (ADAMTSs). Although all the proteases perform ECM protein hydrolysis, this site-specific hydrolysis results in different changes within the scaffold—some alterations lead towards ECM fragmentation, and other alterations lead towards strengthening restructuring [76,77]. Therefore, some proteases stabilize the plaque (like ADAM15), and other proteases destabilize it (such as virtually all MMPs, especially MMP-9) [78,79]. A histological study on human carotid artery specimens has revealed that a more vulnerable plaque phenotype correlated with an increased MMP-14 level and a decreased TIMP-3 (an inhibitor of tissue metalloproteinase) level [80]. Finally, vulnerable plaques are also characterized by neovascularization and proangiogenic factors (such as bFGF or CD137), which are supposed to play a role in plaque destabilization [81,82,83,84]. An optical coherence tomography study on 53 patients has shown an increase in intraplaque neovessel volume in vulnerable and ruptured plaques [85]. In consequence, therapies targeting intraplaque angiogenesis (e.g., axitinib—an inhibitor of VEGF receptor-1, -2, and -3) are considered to be plaque-stabilizing [86,87].

### 3.6. Specific Approaches—Summary

The abovementioned specific approaches are illustrated in Figure 3.

Many different studies have already been conducted to explore specific approaches towards atherosclerotic plaque destabilization. Some examples of such preclinical studies have been presented in the text; others are presented in Table 1.

## 4. An Integrated Approach

Atherosclerotic plaque destabilization is a complex process depending on many different pathways. Therefore, the concept that plaque-stabilizing therapy should target diverse pathways and pathophysiological processes simultaneously is reasonable. For instance, alkaloid berberine has a stabilizing effect on atherosclerotic plaques by the suppression of the ECM regulators MMP9 and EMMPRIN (MMP9 inducer), by autophagy promotion in macrophages, and by promoting antioxidative activity via PPARγ activation (the last effect observed in hyperhomocysteinemia mice) [108,109]. In fact, many agents considered primarily as ‘specifically targeted’ have turned out to be pleiotropic, e.g., lipid-lowering statins or hypoglycemic drugs. As an example, insulin has an anti-inflammatory and plaque-stabilizing effect via the PI3K-Akt pathway, which inhibits the TLR4 MyD88-NF-кB signaling pathway, which was demonstrated using the RAW264.3 monocyte-macrophage lineage [110]. However, the essence of the integrated approach is to target key regulators—transcription factors or common elements of molecular pathways. Accurate examples are statins that inhibit mevalonic acid synthesis (by HMG-CoA blockade), causing subsequent inhibition of isoprenoid intermediates—farnesyl pyrophosphate (FPP) and geranylgeranyl pyrophosphate (GGPP) synthesis. These isoprenoid intermediates serve as lipid attachments necessary for the appropriate activity of small GTPases, especially Ras and Rho, which regulate many cellular pathways. The Rho protein activates Rho kinases (ROCK), which decrease eNOS synthesis and abolish the atheroprotective PI3K-Akt signaling pathway, whereas Rac1 (a member of the Rho subfamily) activates the nicotinamide adenine dinucleotide phosphate (NADPH) oxidase, which is responsible for reactive oxygen species (ROS) generation. As a result, statins induce anti-inflammatory effects (a decrease in proinflammatory cytokines, e.g., IL-6, IL-8, or MCP-1, and adhesive molecule expression, enhancement of Treg and reduction in Th17 lymphocyte differentiation), ECM stabilizing effects (through matrix metalloproteinases reduction), and antioxidative effects (inhibition of ROS generation) [111,112]. Another example of such an integrative approach is the targeting of lipoprotein-associated phospholipase A2 (Lp-PLA2) by its inhibitor, darapladib. Lp-PLA2 generates lysophosphatidic acid, which increases MMP9 production by the NF-кB signaling pathway and simultaneously increases plaque inflammation (by mast cell activation and monocyte recruitment), leading to atherosclerotic plaque destabilization [113,114]. For this reason, Lp-PLA2 inhibition resulted in plaque stabilization in animal models [115]. The aforementioned approaches are addressed towards proteins that regulate diverse molecular pathways, but there is a different option—targeting epigenetic mechanisms driving gene expression. One such method is the usage of RNA interference—proteins are translated from their respective mRNA particles, which can be degraded by RISC complexes, composed of microRNA (miRNA) particles complementary to the corresponding mRNAs. MicroRNAs regulate virtually all stages of atherosclerotic plaque progression by downregulation of the corresponding mRNAs [116]. Some miRNA induces a stabilizing effect on the plaque structure, for instance, miR-520c-3p—a downregulator of RelA/p65 (NF-кB subunit) or miR 181b-5p—a downregulator of NOTCH1 (a promoter of proinflammatory M1 macrophages) [117,118]. In contrast, an example of destabilizing miRNA is miR-124-3p [31,119]. Therefore, an application of agonists (agomirs) of stabilizing miRNAs or antagonists (antagomirs) of destabilizing miRNAs could prevent plaque vulnerability [120]. However, there is another method of an integrative approach promoting atherosclerotic plaque stability—the influence on gene expression through changes in chromatin compaction. DNA strands are wrapped onto nucleosomes built up from histones. Histone acetylation (performed by HATs—histone acetyltransferases) and histone deacetylation (performed by HDACs—histone deacetylases) regulate gene expression making them more or less accessible to RNA polymerase [121]. For instance, HDAC9 increases MMP-1 and MMP-2 expression and toll-like receptor (TLR) signaling at the histone level, and its deficiency or blockade results in inflammation resolution and plaque stabilization [122]. Sirtuins are regulators which also act as histone deacetylases. For example, SIRT2 exerts plaque-stabilizing effects by the inhibition of macrophage polarization towards the M1 phenotype and reduction in iNOS activity. These effects can be induced by Resveratrol, a SIRT2 agonist [123,124]. In summary, an integrative approach (the concept illustrated in Figure 4) makes it possible to have a wide range of effects upon the entire plaque and change its phenotype.

## 5. Clinical Studies Conforming Plaque Stabilization

Numerous preclinical studies have confirmed the plaque-stabilizing effect of different approaches (as mentioned above). However, the promising results of studies on animal models have not been reflected in human clinical studies. There are several explanations for this, i.e., differences in the pathophysiological process of atherosclerotic plaque destabilization (guided by different genes) and in cellular subsets (such as the macrophage population) between laboratory animals and humans, different experimental conditions, and natural diversities in the human population [125,126,127]. Large clinical studies on medicines in the therapy of atherosclerosis particularly concentrate on clinical endpoints—especially mortality, mortality due to cardiovascular reasons, and MACE (a composite endpoint composed of mortality, myocardial infarction, stroke). From this point of view, a canakinumab (anti-IL-1β antibody) given to patients with a previous myocardial infarction (CANTOS study) caused a significant reduction in recurrent cardiovascular events (a dose of 150 mg once a month, relative risk 0.85; 95%CI, 0.74 to 0.98, *p* = 0.021) compared with a placebo [128]. Similarly, colchicine (0.5 mg a day) given to patients after a myocardial infarction caused a significant reduction in composite endpoint (RR 0.77; 95%CI, 0.61 to 0.96, *p* = 0.02) and a marked significant reduction in recurrent myocardial infarction (RR 0.26; 95%CI, 0.1 to 0.7) in comparison with a placebo, which was demonstrated in the COLCOT study [129]. In contrast, a study called STABILITY with darapladib conducted on patients with stable coronary artery disease (without prior myocardial infarction) failed to demonstrate a statistical significance between the darapladib and placebo groups in reference to composite endpoint and mortality, although it showed a slight but significant reduction in major coronary events (9.3% vs. 10.3%, *p* = 0.045) [130]. Similarly, the CETP inhibitor anacetrapib causing an HDL increase demonstrated a slight but significant reduction in major coronary events (REVEAL study) [131]. However, to verify whether a certain type of therapy results in atherosclerotic plaque stabilization, it is necessary to visualize the plaques in the coronary arteries and assess their stability exponents. Methods that enable such visualization are intravascular ultrasound (IVUS) and OCT (optical coherence tomography). Studies in which atherosclerotic plaques were assessed in the light of stability are presented in Table 2 [132].

## 6. Alternative Approaches

This review so far has presented approaches towards atherosclerotic plaque stabilization directed at general mechanisms—the enhancement of stabilizing pathways and the inhibition of destabilizing pathways. In this section, alternative approaches are discussed. As presented above, stable and vulnerable atherosclerotic plaques can transform into one another [15]. Nevertheless, approaches towards the stabilization of concrete and already developed unstable plaque are also under investigation. Examples of such therapies are photodynamic therapy (PDT), plasmonic photothermal therapy (PPTT), cytotoxic chemotherapy, and sonodynamic therapy (SDT) [16,138]. In PDT, administered photosensitizers accumulate within the atherosclerotic plaque (e.g., cross-linked dextran-coated iron oxide (CLIO) nanoparticles which accumulate in macrophages). Then, irradiation of the structure by NIR (near-infrared) causes free radical generation and a cytotoxic effect upon the macrophages (mainly from the M1 subset, dominating in a vulnerable structure), resulting in plaque stabilization. PPTT is quite similar to PDT, but in this method, photoabsorbers (e.g., gold nanoparticles) generate heat when irradiated by NIR. In cytotoxic chemotherapy, special agents are encapsulated in nanomedical liposomes (e.g., prednisolone phosphate) or other formulas such as hyaluronic acid-polypyrrole nanoparticles (e.g., doxorubicin), which make it possible for them to accumulate precisely within the plaque and exert their action. Finally, SDT compounds called sonosensitizers (such as curcumin) localize within the plaque and generate free radicals after exposure to ultrasound. Therefore, SDT is similar to PDT, but ultrasound waves penetrate more deeply than NIR. The principles of the abovementioned methods are illustrated in Figure 5.

Some clinical trials using the abovementioned methods in the therapy of atherosclerotic plaque have already been undertaken [139]. For instance, the NANOM-FIM trial has demonstrated that a group of patients who received silica-gold nanoparticles with subsequent PPTT presented a significantly higher probability of event-free survival than the control group (91.7% vs. 80%) who only received a drug-eluting stent [140]. Moreover, new discoveries such as quantum dots may enable high precision therapy with pre-selected cells introduced into the plaque [141].

In the discussion about approaches towards atherosclerotic plaque stabilization, it must be said that there is also a totally different option—not concentrating on plaque stabilization but rather on the minimization of the consequences of a plaque rupture. The idea of this approach is based on the hypothesis that it is difficult to precisely distinguish between stable and vulnerable atherosclerotic plaque, and microruptures or erosions might occur in many plaques considered to be stable. From that point of view, antithrombotic therapy could be crucial for the prevention of major cardiovascular events such as myocardial infarction because it inhibits clot generation and coronary artery obstruction in the case of a vulnerable plaque rupture or erosion [142]. Although aspirin (a ‘classical’ antithrombotic drug) has turned out to be beneficial in the primary prevention of myocardial infarction (ATT meta-analysis, 12% proportional reduction in serious cardiovascular events per year, *p* = 0.0001), it significantly increased the risk of major gastrointestinal and intracranial bleeding [143]. Therefore, the guidelines (similar to many ESC guidelines) consider aspirin to be a tool for secondary, but not primary, prevention of myocardial infarction.

Finally, approaches towards some pathways in atherosclerotic plaque destabilization still remain controversial, such as targeting pathways involved in plaque calcification (regulated by, for example, oncostatin M). Currently, it is supposed that although plaque macrocalcifications (as in fibrocalcific plaque) increase plaque stability, microcalcifications tend to destabilize the plaque [144].

## 7. Conclusions

In summary, atherosclerotic plaque stabilization is a promising therapy for the reduction in cardiovascular disease burden. Although many interesting discoveries and approaches have already been discovered, there is still a lack of clinically proven methods that enable maintaining the stability of virtually all of the atherosclerotic plaques in a patient. Moreover, many aspects of plaque vulnerability are still controversial and are waiting for a more profound explanation.

## Figures and Tables

**Figure 1 ijms-22-04354-f001:**
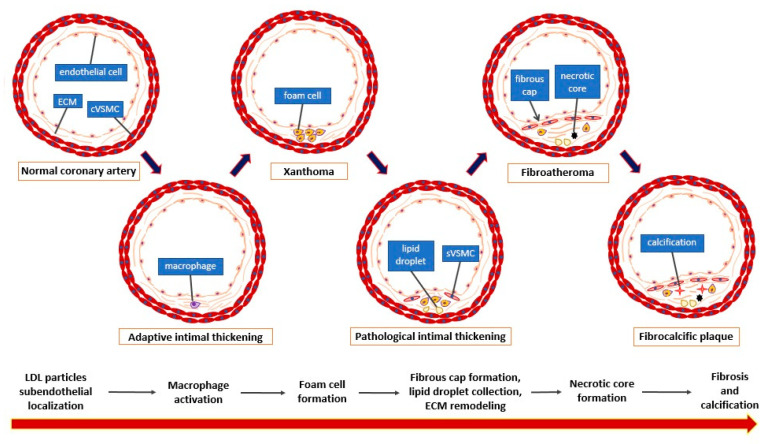
Stages of atherosclerotic plaque development (according to the Virmani classification [6]). cVSMC: vascular smooth muscle cell, contractile phenotype; sVSMC: vascular smooth muscle cell, synthetic phenotype; ECM: extracellular matrix.

**Figure 2 ijms-22-04354-f002:**
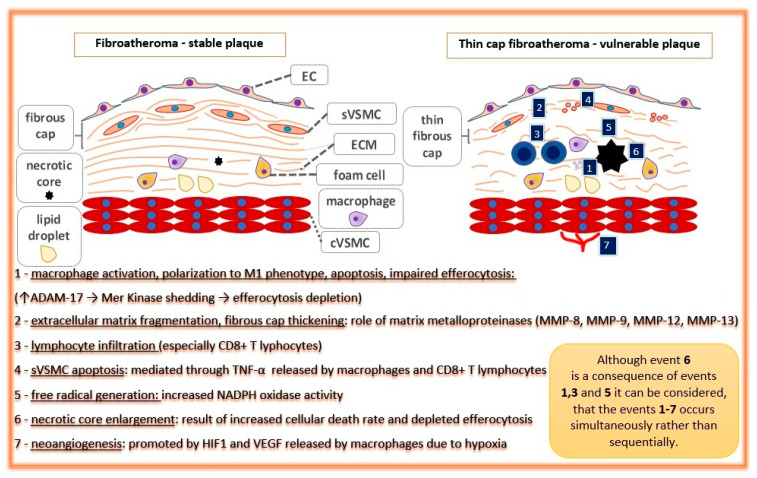
Pathophysiological processes and events leading to atherosclerotic plaque destabilization (events 1–7 in blue boxes). EC: endothelial cell; cVSMC: vascular smooth muscle cell, contractile phenotype; sVSMC: vascular smooth muscle cell, synthetic phenotype; ECM: extracellular matrix.

**Figure 3 ijms-22-04354-f003:**
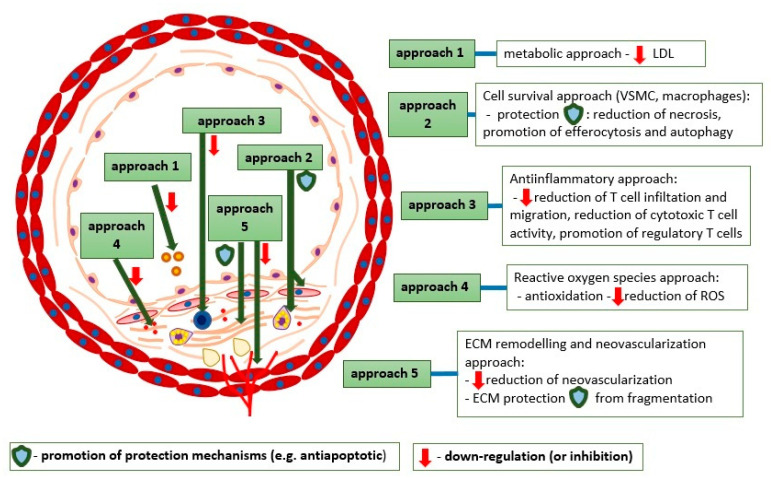
The approaches towards atherosclerotic plaque stabilization therapy and their molecular aspects.

**Figure 4 ijms-22-04354-f004:**
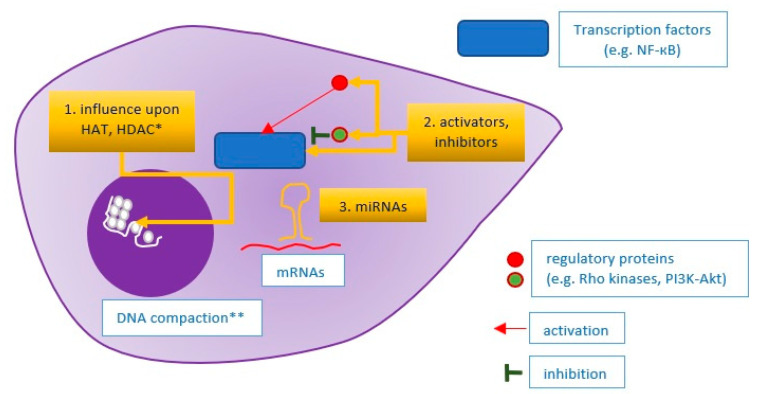
Idea of an integrative approach towards atherosclerotic plaque stabilization. 1. Influence upon mechanisms that are responsible for chromatin compaction and DNA accessibility for transcription machinery; 2. Activation or inhibition of transcription factors, regulating gene expression in the nucleus; 3. Influence upon miRNAs, i.e., particles that inhibit specific mRNA particles in the mechanism of complementarity. * HAT: histone acetyltransferase; HDAC: histone deacetylase; ** DNA compaction relies on nucleosome methylation or acetylation (regulated by HATs and HDACs).

**Figure 5 ijms-22-04354-f005:**
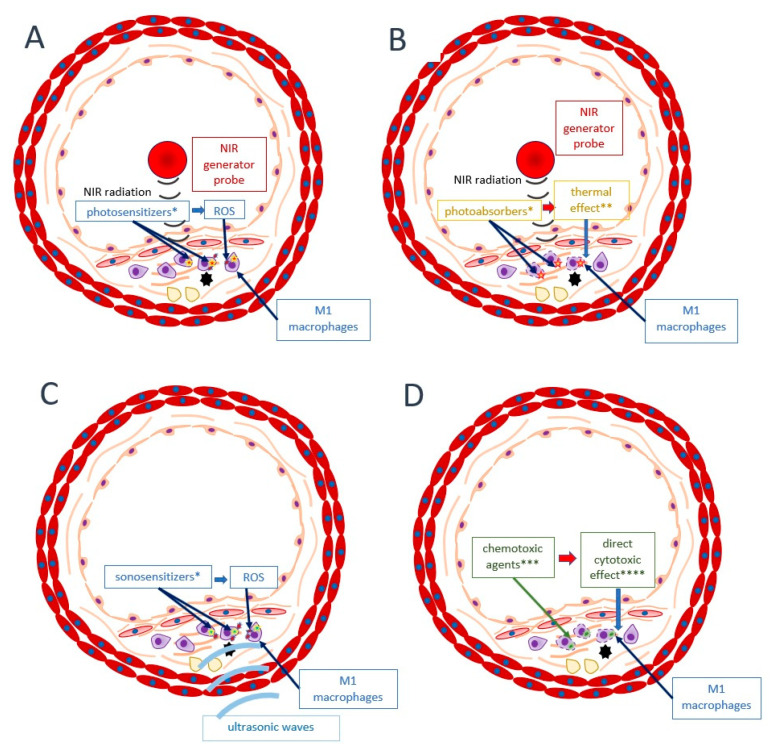
Alternative approaches towards atherosclerotic plaque stabilization targeted at concretely already developed vulnerable plaques. (**A**) photodynamic therapy (PDT); (**B**) plasmonic photothermal therapy (PPTT); (**C**) sonodynamic therapy; (**D**) cytotoxic chemotherapy. * photosensitizer, photoabsorber, or sonosensitizer previously injected into the organism, then accumulated within plaque macrophages; ** effects inducing apoptosis or other forms of cellular death; *** chemotoxic agents encapsulated in nanoformulas and targeted specifically to macrophages; **** apoptosis, necrosis, necroptosis. ROS: reactive oxygen species; NIR: near-infrared.

**Table 1 ijms-22-04354-t001:** Examples of atherosclerotic plaque stabilization treatment (five main approaches) in preclinical models and used drugs.

Approaches	Examples and Mechanisms	Investigated Drugs *
**Metabolic approach**	LDL lowering therapy: (a) Targeting HMG-CoA [21]; (b) Targeting cholesterol absorption [24]; (c) Enhancing LDL clearance [26].	(a) statins (e.g., lovastatin, rosuwastatin, atorvastatin, and pitavastatin); (b) ezetimibe; (c) PCSK9 inhibitors (alirocumab).
	Recombinant HDL particles [28]	75 mg/kg of apoA-I(Milano)
	Iron depletion → increased iron mobilization from macrophages [88,89]	Iron chelation therapy (deferasirox)
	MGL depletion → 2‑AG ↑, CB2R activation [90]	N/A (genetic knock-out organisms used in this experiment)
**Cell survival promotion approach**	(a) Promotion of macrophage autophagy [38]; (b) Efficient efferocytosis of apoptotic bodies [36].	(a) trehalose (disaccharide); (b) AT1R blocker (losartan).
	(a) Inhibition of foam cell formation (e.g., LOX-1 inhibition) [40]; (b) Inhibition of endopasmatic reticulum (ER) stress [91].	(a) Different drugs which caused LOX-1 inhibition (e.g., Resveratrol, tanshinone II-A, and berberine); (b) 4-phenyl butyric acid (PBA)—a chemical chaperone).
	STAT6 upregulation → M2 macrophage polarization [92]	N/A (overexpression by recombinant pcDNA)
	Prevention from excessive PARP1 activation by severe DNA damage → prevention from ATP depletion [93]	For example, 3-Aminobenzamide (3-AB), doxycycline, thieno(2,3-c)isoquinolin-5-one (TIQ-A)
**Anti-inflammatory approach**	(a) Chemokine inhibition (e.g., CCL5 and CXCL10 via TWEAK blockade) [94]; (b) Cell adhesion molecule (e.g., VCAM‑1) inhibition [95].	(a) anti-TWEAK mAb, maraviroc (CCR5 antagonist) [96]; (b) chalcone derivate (1m−6).
	(a) Proinflammatory cytokine (e.g., IL-6, IL‑12, IL-17, IL-18) inhibition [46,97,98]; (b) Anti-inflammatory cytokine (such as IL‑10, IL-37) promotion [55,99].	(a) For example, IL-6 neutralizing antibody (toclizumab); (b) For example, dietary nitrate (L-arginine).
	Cytotoxic CD8 + T lymphocyte (Tc) depletion [100]	CD8α or CD8β targeted monoclonal antibody
	Regulatory T lymphocyte (Treg) promotion [101].	For example, IL-2, mycophenolate mofetil, vitamin D, rapamycin, G-CSF
**Reactive oxygen species approaches**	Downregulation of ROS generators (e.g., NADPH oxidases NOX2) [71]	Nox2 inhibitor peptide (a chimeric 18-amino acid peptide)
	Attenuation of ROS derivative (e.g., 7β‑OH) activity [60].	N/A (indirect methods like conjugation by glutathione)
	Promotion of ROS scavengers (such as HO-1 induced by Nrf transcriptional factor) [102]	N/A
	Direct ROS abruption (e.g., polyphenols) [103]	Different polyphenols (in this study–apple polyphenols)
**ECM remodeling and neovascularization approach**	Inhibition of matrix metalloproteinase (MMPs) synthesis and activity (especially MMP9) [104]	Ghrelin
	Promotion of collagen synthesis (e.g., by melatonin through Akt phosphorylation and subsequent P4Hα1 upregulation) [105]	Dietary nitrate treatment (KNO_3_ or KNO_2_)
	Influence on fibronectin (e.g., blockade of fibronectin-integrin α5 pathway) [106]	In vivo knockdown of phosphodiesterase 4D5 (siRNA)
	Inhibition of neovascularization (e.g., through bFGF blockade) [107]	K5 (a small molecule bFGF-inhibitor)

Abbreviations: MGL: monoglyceride lipase; 2-AG: 2-arachidonoylglycerol (endocannabinoid); PBA: 4-phenylbutyric acid; mAb: monoclonal antibody. * if applicable (in some studies, transgenic organisms were used to investigate specific cellular pathways).

**Table 2 ijms-22-04354-t002:** Clinical studies with outcomes referring to atherosclerotic plaque stabilization visualized with imaging methods (IVUS or OCT).

STUDY NAME	Treatment	No. of Investigated Patients (Period)	Clinical Outcome (MACE, Mortality)	Plaque Stabilization Effect (IVUS or OCT)
**GAIN** [133] **^1^**	**Atorvastatin (20–80 mg)** vs. placebo	65 and 66 (12 months)	Any ischemic event: **21.5%** vs. 31.8% (*p* = 0.184)	IVUS: Larger hyperechogenicity index **42.2%** vs. 10.1%, *p* = 0.021
**REVERSAL** [134] **^2^**	**Atorvastatin 80 mg (intensive lipid-lowering)** vs. Pravastatin 40 mg(moderate lipid-lowering)	253 and 249 (18 months)	Death: 0.3% vs. 0.3%—NS Myocardial infarction: 1.2% vs. 2.1%—NS	IVUS: Lower percent atheroma volume change **0.2%** vs. 1.6%, *p* < 0.001
**PRECISE-IVUS** [37] **^3^**	**Atorvastatin * + ezetimibe (10 mg)** vs. Atorvastatin * alone	102 and 100 (9–12 months)	Cardiovascular events ** 11% vs. 14%—NS	IVUS: Change in normalized TAV −6.6% vs. −1.4%, *p* < 0.001
**GLAGOV** [135] **^4^**	**Statin *** + PCSK9i (evolocumab 420 mg monthly)** vs. Statin *** alone	423 and 423 (19 months)	Death: **0.6%** vs. 0.8%—NS Non-fatal myocardial infarction: **2.1%** vs 2.9%—NS	IVUS: Change in TAV **−5.8%** vs. –0.9%, *p* < 0.001
**Christoph et al.** [136]	**Pioglitazone (30 mg)** vs. Placebo	27 and 27 (9 months)	Insignificant differences, no MACE registered	VH-IVUS: Decrease in the necrotic core **−1.3%** vs. + 2.6%, *p* = 0.008
**Tondapu et al.** [137]	**Rosuvastatin (10 mg)** vs. Atorvastatin (20 mg)	24 and 19 (12 months)	Not applicable	OCT: Increased FCT **** **171.5** vs. 127.0 μm, *p* = 0.03; Decreased macrophages

^1^ German Atorvastatin Intravascular Ultrasound Study Investigators; ^2^ Reversal of Atherosclerosis with Aggressive Lipid Lowering Study; ^3^ Plaque Regression With Cholesterol Absorption Inhibitor or Synthesis Inhibitor Evaluated by Intravascular Ultrasound; ^4^ Global Assessment of Plaque Regression With a PCSK9 Antibody as Measured by Intravascular Ultrasound. * Atorvastatin dose adequate for efficient lipid-lowering with target LDL < 70 mg/dL; ** Cardiovascular events—mainly revascularizations of de novo lesions, no cases of death during follow-up in this study; *** Statin dose adequate for efficient lipid lowering with target LDL < 80 mg/dL (or <60 mg/dL, in case of additional risk factors); **** similar baseline fibrous cap thickness in both groups (61.4 μm in rosuvastatin group and 60.8 μm in atorvastatin group); IVUS: intravascular ultrasound; VH-IVUS: virtual histology-intravascular ultrasound; OCT: optical coherence tomography; TAV: total atheroma volume; FCT: fibrous cap thickness.

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
