# Peer review of "Different Approaches in Therapy Aiming to Stabilize an Unstable Atherosclerotic Plaque"

_ijms, 2021, doi:10.3390/ijms22094354_

Round 1

Reviewer 1 Report

The review “Different approaches in therapy aiming to stabilize an unstable atherosclerotic plaque” by Kowara et al. suggest the idea of atherosclerotic plaque stabilization from the viewpoints of regulation of metabolism, macrophages and cellular death, inflammation, reactive oxygen species, and extracellular matrix remodeling based on the accumulating informations.

This reviewer appears to suggest a clinical strategies against unstable atherosclerotic plaques from the authors’ own views. The manuscript has been well prepared with clear illustrations. Therefore, I believe that this review paper is suitable for publication in IJMS in the present style.

Author Response

We are very grateful for the review. 

Reviewer 2 Report

The authors Michal Kowara , Agnieszka Cudnoch-Jedrzejewska did great job. I only have one concern. Please improve the quality of figures.

Author Response

We are very grateful for the review. According to Your recommendation, we have adjusted the figures and improved their quality (mainly by text font increase). 

Reviewer 3 Report

In a review by Kowara et al. authors have presented different approaches to stabilize an unstable atherosclerotic plaque, such as the regulation of metabolism, the influences on macrophages and cellular deaths mechanism, inflammation and immune reaction, reactive oxygen species, as well as extracellular matrix remodelling and neovascularization. Please find my comments below:

  1. Authors discussed different approaches towards stabilization of atherosclerotic plaque, such as promotion of autophagy and efferocytosis, macrophages polarization, inhibition of NADPH oxidases and so on and so forth. However, in many cases they did not include the names of corresponding drugs, e.g. NOX2 inhibitors or autophagy activators, just to list a few examples, which will be highly helpful for readers.
  2. Figures should be enlarged and better readable, especially captions – they are too small.
  3. Please add detailed description to every Figure legend, especially Figure 4 is unclear without description.
  4. Rearrange Table 1 in a more readable manner, e.g. examples of atherosclerotic plaque stabilization treatment should be in rows. List the names of appropriate drugs, too.

Author Response

We are very grateful for the insightful comments. 

  1. We have intorduced drug names (if applicable) to the manuscript, especially to the Table 1.
  2.  We have improved the figures in the manuscript. 
  3. We have adjusted the figure captions. 
  4. Table 1 has been deeply rearranged, now it's divided to 3 columns - approaches, mechanisms and investigated drugs. 

Round 2

Reviewer 3 Report

I don't have further comments.